# Microstructural Evolution and Mechanical Properties of Friction Stir Welded Butt Joints of 5A06 Alloy Ultra-Thin Sheets

**DOI:** 10.3390/ma12233906

**Published:** 2019-11-26

**Authors:** Yang Han, Xiaoqing Jiang, Tao Yuan, Shujun Chen, Dongxiao Li, Zhenyu Qi

**Affiliations:** 1Engineering Research Center of Advanced Manufacturing Technology for Automotive Structural Parts, Ministry of Education, Beijing University of Technology, Beijing 100124, China; hanyang071@163.com (Y.H.); xjiang@bjut.edu.cn (X.J.); sjchen@bjut.edu.cn (S.C.); bjutqzy@163.com (Z.Q.); 2Beijing Spacecrafts Manufacturing Factory, Beijing 100080, China; azure0117@126.com

**Keywords:** friction stir welding, 5A06, thin plate, dynamic recrystallization, mechanical properties

## Abstract

Ultra-thin plates have great potential for applications in aircraft skin, the packaging industry, and packaging of electronic products. Herein, 1 mm-thick 5A06 Al alloy was welded with friction stir welding. The microstructural evolution of the welds was investigated in detail with optical microscopy, scanning electron microscopy, and electron backscatter diffraction. The results showed that the friction stir welds of 1 mm-thick 5A06 Al alloy were well formed without obvious defect and with a minimum thickness reduction of 0.025 mm. Further, the grain size and the proportion of low-angle grain boundaries decreased with decreasing welding speed, because of the increasing degree of dynamic recrystallization. Among all of the welded joints, the welding speed of 100 mm/min yielded the smallest grain size and the highest proportion of high-angle grain boundaries, and thus the best mechanical properties. Specifically, the tensile strength of the joint was greater than that of the base material, while the elongation reached 80.83% of the base material.

## 1. Introduction

Aluminium alloys are widely used in the aerospace, transportation, shipping, and chemical industries owing to their good mechanical properties such as high specific strength, high specific stiffness, low density, high strength ratio, and good plasticity [1,2,3]. In particular, the use of aluminium alloys for vehicle weight reduction has increased significantly in recent years [4]. Further, the application of aluminium alloy thin plate is significant, being widely used in aircraft skin, the packaging industry and the packaging of electronic products [5]. The joining methods of aluminium alloys mainly include clinching, self-piercing riveting, resistance spot welding, arc welding, and friction stir welding (FSW) [3,6,7]; where welding technologies are typically used to realize the sealed joining of the aluminium alloy. However, thin plates of Al alloys are difficult to weld by arc welding owing to the defects that typically occur during welding, such as pores and hot cracks [8]. Furthermore, it is difficult to control the deformation of the thin plates during welding. Essentially, it is difficult to obtain a high-quality joint of a thin plate of aluminium alloy using conventional fusion welding [9,10].

The solid-state welding method of FSW was developed in 1991 by The Welding Institute, UK, to resolve these problems. The FSW of thin metal plates has been widely studied by researchers all over the world [11,12,13]. Ahmed et al. [14] designed a fixture for aluminium alloy sheets that effectively reduced the sheet deformation and heat loss, and the joining efficiency reached 74%. Huang et al. [5,15] studied the effect of the pin shape on the joint properties. Rodrigues et al. [16] found that the grain size of joints produced via a conical shoulder was larger than that produced with a scrolled shoulder, while the size of the precipitates became smaller. Scialpi et al. [17] joined 0.8 mm 2024-T3 and 6082-T6 Al by FSW, obtaining a maximum tensile strength of the FSW joint that was 69% of that of 6082-T6 Al. Chen et al. [18] welded 1.4 mm 2A97 Al, and concluded that the grain size in the stir zone (SZ) and the thermo-mechanically affected zone (TMAZ) is much smaller than that of the base material (BM). Li et al. [19] researched the FSW of 1 mm 2024-T4 Al, and found that a large number of low-angle grain boundaries (LAGBs) existed in the SZ that helped to improve the mechanical properties of the joints.

However, research on the FSW of ultra-thin Al–Mg alloys is limited, though research focusing on the FSW of thick Al–Mg alloys could provide useful information for ultra-thin Al–Mg alloys. Rodrigues et al. [20] obtained suitable welding parameters for the FSW of the AA5083-H111 Al alloy 4 and 6 mm thick. Guo et al. [21] welded 108 mm-thick 5A06 Al alloys and found that the grain size of equiaxed grains decreased gradually along the thickness direction. Chen et al. [22] reported that recrystallization texture components were replaced by shear texture components with increasing rotation speed in the FSW of 5.17 mm-thick 5A06 Al alloys. Most of these investigations were focused on the FSW of thick Al–Mg plates, while the FSW of ultra-thin plates is different from that of thick plates in the following ways: (1) the ultra-thin plate is easily deformed during FSW, and the deformation during the welding process can seriously affect the weld quality; and (2) ultra-thin plates exhibit rapid heat dissipation during welding, and thus the temperature distribution and microstructural evolution in thickness direction are different from that of thick plates. In fact, Sattari et al. [23] studied the effect of welding speed on the defects of 0.8 mm-thick 5083 Al alloy during FSW. Most previous studies, however, have mainly focused on the performance of the welding process while varying the welding parameters, and the microstructural analysis of the welded joints has been less reported. Thus, it is beneficial to study the FSW of ultra-thin Al–Mg alloy and investigate the microstructural evolution.

In the present study, 5A06 Al alloys 1 mm thick were butt welded via FSW, whereupon the microstructural and texture evolution were investigated in detail by optical microscopy, scanning electron microscopy (SEM), and electron backscatter diffraction (EBSD). To study the effect that the small thickness has on the weld properties, the microstructural evolution along the horizontal and vertical directions were analyzed. Finally, the effect of the welding parameters on the microstructure and mechanical properties were discussed.

## 2. Materials and Methods

The material used in this research was rolled-then-annealed 5A06 Al with a size of 300 mm × 100 mm × 1 mm, whose surface oxide was removed before welding. The welding tool was made of H13 steel with a shoulder diameter of 8 mm, and whose pin was characterized by concentric grooves. The 0.8 mm-long pin tapered from 2.6 mm at the top to 3 mm at the bottom. The welding parameters were chosen with reference to previous research and the authors’ preliminary study. The tool tilt angle and plunge depth were respectively chosen on the basis of the research results of Rodrigues et al. [22] and Huang et al. [5]. In addition, according to preliminary studies, the weld quality is sensitive to the rotation speed of the tool, where the ultra-thin plate is prone to bulging with a high rotation speed and to forming groove defects with a low rotation speed. Thus, the rotation speed herein was 1000 rpm and was held constant during variation of the welding speed. This work focused mainly on the microstructural evolution of an ultra-thin Al–Mg alloy. Therefore, the welding parameters were held constant except the welding speed, because the contribution of the welding speed to the welding efficiency is greater than that of the rotation speed. The welding parameters are shown in Table 1. The welding direction was parallel to the rolling direction of the plate and stainless steel was used as the backing plate.

After welding, the specimens for microstructural analysis were prepared by the standard metallography procedure and etched with a solution of 2 mL hydrofluoric acid, 5 mL nitric acid, 10 mL hydrochloric acid, and 90 mL distilled water. The microstructure of the joint was analyzed using optical microscopy (LEXT OLS 4100, Olympus Corporation, Tokyo, Japan). The specimens were prepared for EBSD via electrolytic polishing with 30% nitric acid and 70% alcohol, and the EBSD voltage was 12 V at a temperature of 253 K.

The hardness was measured using a Vickers hardness tester (SCTMC DHV-1000Z, Shanghai Optical Instrument Factory, Shanghai, China) with a 200 g load for 15 s. The measuring point distribution is shown in Figure 1, and the step length between measurements was 0.35 mm. Tensile tests were performed at room temperature under the ASTM-E8 standard, using a tensile velocity of 3 mm/min. The tensile strength value was taken as the average value of three samples produced with the same welding condition. After the tensile test, the fracture surface was analyzed using scanning electron microscopy (SEM; FEI Quanta 200, FEI, Eindhoven, Holland).

## 3. Results and Discussion

### 3.1. Surface Appearance and Joint Cross-Section Macrographs

Figure 2 shows the welds produced by different welding speeds, where the surface can be seen to be smooth without grooves and ploughs. As shown in Figure 2a, a large flash exists when the welding speed was 100 mm/min. With increasing welding speed, the flash was observed to decrease gradually, which is in agreement with a previous report [15].

Figure 3 shows the cross-section microstructure of the welds produced by different welding speeds, where no void defects were found. Observing the microstructure, the weld cross-section could be divided into the BM, the heat affected zone (HAZ), the TMAZ and the SZ. It was found that the width of the SZ gradually decreased with increasing welding speed, which was the result of a decreasing heat input. This result is in accordance with the research reported by Huang et al. [5,15]. The SZ was “basin-shaped” at the welding speeds of 100 and 150 mm/min, while the SZ became “bowl-shaped” when the welding speed increased to 200 mm/min. However, a black line was observed at the root of the SZ, which is located at the tip of the pin. This line may be caused by an inconsistency of material flow direction between the upper and lower sides of the pin tip.

In the FSW of thin plates, thickness reduction of the joints typically plays an important role in the mechanical properties. This is because a stress concentration can easily form at the edge of the shoulder affected zone as a result of the thickness reduction. Therefore, it is necessary to study the effect that the welding parameters have on the thickness reduction of the joints.

To simplify the analysis, this work used the thickness reduction rate to evaluate the thickness reduction, given as:*R* = (1 − *t_sz_*/*t_bm_*) × 100%(1)
where *R* is the thickness reduction rate; and *t_sz_* and *t_bm_* are the thicknesses of the SZ and the BM, respectively.

The effect of the welding speed on thickness reduction is shown in Figure 4, where the thickness reduction rate of the joints decreased with increasing welding speed. The thickness reduction rate was 5.74% (0.0528 mm) at a welding speed of 100 mm/min, and was only 2.53% (0.0233 mm) at a welding speed of 200 mm/min. With increasing welding speed, the degree of material softening decreased as the heat input reduced, where a decreased degree of material softening increases the difficulty for material flow as driven by the shoulder. Meanwhile, the number of tool rotations per unit distance decreases, which reduces the number of times that the shoulder will drive the materials at both the advancing side (AS) and the retreating side (RS). This, in turn, contributes to a gradual decrease of the thickness reduction rate of the joints with increasing welding speed.

### 3.2. Microstructural and Texture Evolution of FSWs

Figure 5 shows the inverse pole figure color map and the corresponding color key in the contours of the BM. Herein, the transverse (TD) and normal (ND) directions represent the horizontal and vertical directions, respectively, and the welding direction (WD) was parallel to the rolling direction of the BM. The average grain size of the BM was 10.33 ± 3.37 µm, based on the linear intercept method. According to the inverse pole figure color map (Figure 5a), the grain orientation distribution of the BM was uniform. Meanwhile, as shown in the color key map (Figure 5b), the texture of the BM mainly consisted of {111}, {001}, and {112}, and the maximum texture intensity (Imax) was 1.18.

#### 3.2.1. Effect of Welding Speed on Weld Microstructure

Figure 6 shows the microstructure of the SZ for different welding speeds, where the SZ was mainly composed of refined equiaxed grains. The average grain size of the SZ at the welding speed of 100, 150, and 200 mm/min was 3.98 ± 1.12, 4.77 ± 1.51 and 5.17 ±1.66 µm, respectively, according to the results obtained by the linear intercept method. Thus, the grain size gradually increased with the welding speed. We note that the equiaxed grains of the SZ are primarily formed by dynamic recrystallization, which is mainly affected by the temperature and deformation during welding. Yu et al. [24] reported that dynamic recrystallization could be accelerated by rapid heating and a high strain rate. Thus, a more rapid temperature rise or a larger deformation will produce a more complete dynamic recrystallization, and consequently a smaller grain size. At a low welding speed, the dwell time is longer and the number of rotations is larger, causing a higher strain rate of the material and a greater heat input. Thus, the strain rate caused by tool rotation and the heat input at a welding speed of 100 mm/min was greater than that at a welding speed of 150 or 200 mm/min. Therefore, the dynamic recrystallization was faster and more complete, and the grain size was smaller at the lower welding speed.

In addition, the grain size deviation in the SZ was 1.12 µm at a welding speed of 100 mm/min, increasing to 1.51 and 1.66 µm at welding speeds of 150 and 200 mm/min, respectively. The degree of dynamic recrystallization decreased with increasing welding speed, so the degree of dynamic recrystallization was not uniform in the different welds. In other words, the grains sizes across the weld length and along the weld thickness were heterogeneous for high welding speeds. Thus, the grain size deviation increased when the welding speed increased from 100 to 150 and 200 mm/min. Further, Luo et al. [25] reported that grain growth is a process in which large grains grow at the expense of their small neighbours, which leads to a reduction of grain size deviation as the fine grains are consumed via grain growth. Low welding speeds induce longer high temperature dwell times, providing more time for the grains to grow. Thus, the grains may have had more time to grow at the welding speed of 100 mm/min than at 150 and 200 mm/min, causing a smaller grain size deviation of the former weld because more fine grains could be consumed. It should be noted that the heat dissipation of the ultra-thin plate is fast, so the primary factor affecting the grain size deviation in this work was the degree of dynamic recrystallization and not the high temperature dwell time. In other words, the grain growth rate may be stronger during the high temperature dwell time in the weld with a 100 mm/min welding speed, but the grains could not grow bigger than those in the weld with a 150 or 200 mm/min welding speed.

Figure 6d–f show the inverse pole figure color maps of the SZ at the welding speeds of 100, 150, and 200 mm/min, where the maximum texture intensity was 1.283, 1.228, and 1.320, respectively; and the textures were {212}, {101}, and {111}, respectively. During FSW of Al alloys, the SZ undergoes drastic shear deformation, and thus a shear texture is formed in the SZ. However, the texture intensity decreases owing to the occurrence of dynamic recrystallization, which is induced by shear deformation and high temperature. Thus, the texture intensity of the SZ in this work was slightly stronger than that of the BM. Actually, the texture intensity difference between the SZ and the BM will be affected by both the material type and the welding parameters. Herein, the maximum texture intensity in the SZ did not differ significantly among the welds produced with varying welding speed. This signifies that, at a rotation speed of 1000 rpm, increasing the welding speed from 100 to 200 mm/min has little effect on the texture intensity.

Figure 7 shows the misorientation angle distribution in the SZ at different welding speeds. With decreasing welding speed, the proportion of LAGBs decreased gradually. At the welding speed of 100 mm/min, the proportion of high-angle grain boundaries (HAGBs; i.e., grain misorientation angle greater than 15°) reached 85.39%, which is owing to the high heat input and the relatively complete dynamic recrystallization. Jazaeri et al. [26] reported that LAGBs are primarily caused by dynamic recovery during deformation, and are mainly within large grains or along grain boundaries; while HAGBs are formed at the boundary of two grains because of dynamic recrystallization, plastic material flow and the welding thermal process [18]. Song et al. [27] reported that material fluidity can be improved by a higher heat input, which can promote complete dynamic recrystallization. During FSW, the heat input increased with decreasing welding speed, and thus the proportion of HAGBs with the 100 mm/min welding speed was higher than that with the 150 and 200 mm/min welding speeds in this work owing to the complete dynamic recrystallization induced at the low welding speed (i.e., high heat input). According to the results shown in Figure 6 and Figure 7, the grain size of the weld with the welding speed of 100 mm/min was smaller than that with 150 and 200 mm/min welding speeds, while the texture intensity was weaker and the proportion of HAGBs was higher. This suggests that a higher hardness and tensile strength exists in the weld produced with a 100 mm/min welding speed, which may contribute to it exhibiting the best performance of the joints studied in this work.

#### 3.2.2. Joint Microstructural and Texture Evolution along the Vertical Direction

The microstructural evolution at the top, centre, and the bottom of the SZ of the weld produced with a welding speed of 150 mm/min is shown in Figure 8, where the average grain size of these three zones was 4.05 ± 1.19, 4.77 ± 1.51, and 5.25 ± 1.69 µm, respectively. The average grain size in the top zone was the smallest because the heat production was the highest in this zone. According to Su [28], the heat production via friction between the shoulder and workpiece is more than 80% of the total heat production during FSW, because the contact area between the shoulder and workpiece is the greatest. In addition, the heat production rate and strain rate caused by the tool gradually decreased herein from the top to the bottom because of the conical shape of the pin. However, a high heating rate and high strain rate are beneficial for accelerating the dynamic recrystallization. Thus, the dynamic recrystallization in the bottom zone was less complete than in the top zone, leading to the observed increase of average grain size from the top to the bottom. It should be noted that the workpiece is very thin and the thermal conductivity of aluminium alloy is good, so the effect of the temperature difference along the vertical direction is relatively small. Thus, the difference of strain rate from the top to the bottom may play an important role on the dynamic recrystallization. Jiang et al. [29] reported that a high strain rate can lead to an enhanced pile-up of dislocations and a higher stored energy, which can enhance nucleation and facilitate the process of dynamic recrystallization.

As shown in Figure 8d–f, the texture of the top and bottom zones of the SZ was {001} while that of the centre zone was {101}. Further, the Imax of the top, centre, and bottom zones were 1.495, 1.228, and 1.673, respectively. The texture intensity of the SZ was larger than that of the BM, which indicates that the welding parameters adopted in this work increased the concentration of the grain orientation distribution in the SZ. Meanwhile, no significant change in the texture intensity from the top to the bottom zone was observed, indicating that within a single joint the texture changed little in the vertical direction of the thin plate.

The misorientation angles of the top, centre and bottom of the SZ are shown in Figure 9, where the proportion of LAGBs in these three zones were 17.76%, 14.61%, and 12.41%, respectively. Generally, the grain is small when the dynamic recrystallization is complete, and thus the proportion of LAGBs is low. However, in this work, the proportion of LAGBs decreased gradually from the top to the bottom of the SZ, while the average grain size increased gradually. During the FSW of ultra-thin aluminium plates, the effect of the temperature gradient along the vertical direction of the SZ can be ignored owing to the good thermal conductivity and the minimal thickness of the ultra-thin aluminium plate. The grain deformation generates more dislocations and the frictional heat promotes the movement and redistribution of these dislocations, and therefore the LAGB is primarily related to the change of dislocations [18]. From the top to the bottom of the SZ, the degree of grain deformation gradually decreased, indirectly resulting in a reduction of the proportion of the LAGBs.

#### 3.2.3. Joint Microstructural and Texture Evolution along the Horizontal Direction

Figure 10 shows the microstructure at different locations in the horizontal direction of the joint. The average grain size of the HAZ was larger than that of the BM in both the advancing side (AS) and the retreating side (RS) owing to the influence of the thermal cycle. In the TMAZ, the material was affected by the thermal cycle as well as plastic deformation via the tool rotation. Thus, though dynamic recrystallization occurred in the TMAZ, it was incomplete because the heat and deformation were insufficient. Therefore, the average grain size of the TMAZ was smaller than that of the BM, and the average grain size of RS-TMAZ was slightly smaller than that of AS-TMAZ. The grain size of the SZ was fine and uniform owing to the complete dynamic recrystallization induced by severe deformation and high temperature. Further, the grain size of the AS-SZ was obviously larger than that of the RS-SZ, while the grain size distribution of RS-SZ was relatively uniform. The grain size of the AS was larger than that of the RS in both the SZ and the TMAZ, owing to the higher temperature at the AS than that at the RS [30]. In particular, the tool rotation direction in the AS is the same as the welding direction, making the relative friction rate between the AS material and the tool higher than that of RS. This induces more frictional heat in the AS, and higher temperatures promote grain growth. The proportion of LAGBs, the grain size and the Imax of the different zones in the joint are shown in Figure 11. From the RS-HAZ to the AS-HAZ, the average grain size and grain size deviation initially decreased and then increased. Compared with the other regions in the joint, the average grain size and grain size deviation of the SZ centre were the smallest owing to its complete dynamic recrystallization.

The texture intensity of the weld was slightly greater than that of the BM (Figure 10a1–g1 and Figure 11), where an increase of texture intensity indicates an increase of the orientation degree of the grains. The increments of increase are not large, however, indicating that the texture has little effect on the comprehensive mechanical properties of the welded joints. The dominant texture in the HAZ was {212}, while that of the TMAZ was {101}; and various texture types were identified in the SZ, including {433}, {101}, and {001}. This variation of texture is primarily related to the material fluidity and influence of heat at different locations in the welded joints. From the RS-HAZ to the SZ the texture intensity gradually decreased, reaching its lowest value of 1.228 in the centre of the SZ. However, from the AS-HAZ to the SZ the texture strength did not decrease gradually. The maximum texture strength of 2.275 was found at the AS-SZ, indicating that the grains were concentrated in {001}. Further, this shows that the anisotropy of the AS-SZ is strong and that the AS-SZ possesses the weakest texture strength.

Comparing the proportion of LAGBs at different positions of the joints (Figure 11), it was found that the proportion of LAGBs at the SZ centre was greater than that of the BM. Li et al. [15] also obtained this same result, which was attributed to the production of a large amount of sub-grain structures in the SZ. However, in the horizontal direction of the joint, the proportion of LAGBs at the SZ centre was the lowest, which is because the large deformation and high heat input in the SZ centre induces a more complete dynamic recrystallization. Hu et al. [31] reported that, with increased heat input, the deformation-induced LAGBs migrate to form sub-structural grains. If the heat input and welding time are sufficient, these sub-structural grains can aggregate under the effect of dynamic recovery and grow to form recrystallized grains, whose grain boundaries become HAGBs [26,31]. From the SZ centre to the TMAZ, the proportion of LAGBs gradually increased. This is also because the degree of dynamic recrystallization reduced from the SZ to the TMAZ, which reduced the number of recrystallized grains.

In conclusion, the results of grain size, texture intensity, and distribution of grain boundaries indicate that the AS of the joint is a weak zone.

### 3.3. Mechanical Properties of the Joints

Figure 12 plots the hardness of the joints produced with varying welding parameters, where the hardness distribution was basically consistent with the cross-section morphology of the welds. The hardness clearly decreased from the top to the bottom of the SZ, which shows good agreement with the results of Zhao et al. [32]. This is mainly because the deformation at the bottom of the SZ was weaker than at the top, and thus the degree of dynamic recrystallization gradually decreased from the top to the bottom of the SZ, and thus the grain size gradually increased. According to the Hall–Petch relationship, larger grain sizes exhibit reduced hardness values [33]. In addition, the proportion of LAGBs decreased from the top to the bottom of the SZ, which also causes a reduced hardness [15]. The comparative hardness of the areas within the joint was SZ > TMAZ, which was mainly affected by the grain size. It is worth noting, however, that the HAZ hardness was greater than that of the BM. This is owing to the fact that there was little grain size difference between the HAZ and the BM, while the proportion of LAGBs in the HAZ was much higher than that in the BM. Further, more LAGBs indicate a higher dislocation density, and thus a higher hardness value [15]. There was no obvious difference in the hardness of the AS and the RS of the joints, mainly because of the limited difference in grain sizes and the proportion of LAGBs.

As shown in Figure 13, the average hardness of the joints at the welding speeds of 100, 150, and 200 mm/min were 87.34, 86.49 and 86.12 HV, respectively, which are all higher than that of the BM. It should be noted that hardness is affected by grain size in the present work. It is seen in Figure 6 and Figure 13 that, with the increase of welding speed, the grain gradually became coarser and the average hardness of the joints gradually decreased. The high hardness area of the joint decreased gradually with increasing welding speed, as shown in Figure 12. With constant rotation speed, the heat input decreased with increasing welding speed, which caused a decrease in the SZ area. Further, the hardness of the SZ was greater than that of the BM owing to refined grains. In addition, the hardness of the weld bottom decreased according to Figure 12 because the heat input was insufficient to promote dynamic recrystallization of the materials at the bottom of the plate. Actually, because of the extreme thinness of the plate, the heat can be easily spread to the bottom of the plate, but we note that the heat can be easily diffused owing to the minimal thickness.

The mechanical properties of the joints and the BM are given in Figure 14. It can be seen that, with increasing welding speed, the tensile strength of the joints initially decreased and then increased. The elongation of the joints exhibited the same variational trend. The position of the fracture in the joint produced with a 100 mm/min welding speed was located in the BM. Generally speaking, the fracture of tensile samples occurs in the weakest sample position, signifying that the tensile strength of joint area was higher than that of the BM in this work. In addition, both the tensile strength and elongation of the 100 mm/min weld shown in Figure 14 were close to the properties of the BM. However, when the welding speed was increased to 150 and 200 mm/min, the fracture occurred in the centre of the SZ and the tensile strength reached 85% and 95% of the BM, respectively. The decrease of the tensile strength with increasing welding speed is primarily caused by remnants of the oxide layer in the weld [18]. Besel et al. [34] reported that, although the original oxide layer is removed prior to welding, a new oxide layer forms immediately under ambient lab conditions. When the welding speed is low (e.g., 100 mm/min), the tool has more rotations per unit distance, so the materials can be rotated completely and the oxide layer can be distributed uniformly. Furthermore, joints produced at the welding speed of 100 mm/min exhibited the smallest grain size and the highest proportion of HAGBs among all of the joints, as shown in Figure 6 and Figure 7, where finer grains and higher proportion of HAGBs are beneficial for increasing the tensile strength of the joint. It is noteworthy that the tensile strength of the joints produced with a 200 mm/min welding speed is higher than that produced with 150 mm/min, which is mainly owing to the decreased thickness reduction ratio of the joints with increasing welding speed. The thickness reduction ratio of the joints produced with a 150 mm/min welding speed was 3.56%, while that with 200 mm/min was 2.53%. Thus, both the oxide layer and thickness reduction play an important role in the mechanical properties of the FSW-processed ultra-thin Al–Mg alloys.

The fracture surface of the BM and the welds produced with various welding speeds are shown in Figure 15 and Figure 16. The fracture morphology of the joint produced with a welding speed of 100 mm/min (Figure 15) was different from those produced with the welding speeds of 150 and 200 mm/min (Figure 16) owing to the different fracture locations. The fracture morphology of the joint produced with a welding speed of 100 mm/min was similar to that of the BM because the fracture zone of the joint was located at the BM. As shown in Figure 15, the necking occurred during the fracture of the BMs, and numerous tearing ridges and dimples were present in the fracture surface. Therefore, it is concluded that the fracture mode of the BM is a typical ductile fracture.

As shown in Figure 16a,b, four different layers were present on the fracture surface from the top to the bottom of the joints produced with welding speeds of 150 and 200 mm/min, labelled herein as layer-I, layer-II, layer-III, and layer-IV. The high-magnification details of layer-I are shown in Figure 16c, where the morphology of the layer-I surface was granular with a few dimples and tearing ridges, indicating a mixed fracture mode including both brittle and ductile fracture. An arc-shaped fracture interface was seen in layer-II, shown in Figure 16d, where an intergranular fracture mode was obvious. The detailed surface image of layer-III, shown in Figure 16e, exhibited many tearing ridges and dimples, indicating that layer-III was a ductile fracture layer. Finally, layer-IV was actually a weak junction area located at the root of the SZ, as shown in Figure 16f. The particles present in layer-IV were relatively flat and long, and were similar to the typical rolling grain characteristics of the BM. It is therefore concluded that the fracture mode of the faster-welding-speed joints was a mixture of intergranular brittle fracture and ductile fracture. Chen et al. [18] found that the oxide layer remnants in the centre of the SZ is the primary reason for the delamination of the fracture surface. Furthermore, the dimples in the joint produced with a welding speed of 200 mm/min were larger and deeper than those in the joint produced with a welding speed of 150 mm/min, which is one of the reasons for the high strength and elongation of joints produced at 200 mm/min.

In conclusion, the bottom of the joint and the remnants of the oxide layer in the SZ are the weak areas of the FSW-produced ultra-thin plate joint, which seriously impact the mechanical properties of the joint. Further, this study shows that the effect of the oxide layer remnants on mechanical properties of the joints can be reduced by decreasing the welding speed.

## 4. Conclusions

In this paper, the 1 mm-thick 5A06 Al alloy was welded by the FSW process, whereupon the microstructural evolution of the welds produced with varying welding speeds were investigated and the effect of the microstructure on the mechanical properties was discussed. The main conclusions are as follows:No obvious surface defects were found in the FSW joints of the 1 mm-thick 5A06 Al alloy. The thickness reduction of the joint decreased with increasing welding speed, where the smallest thickness reduction was 0.025 mm.The tensile test fracture occurred in the BM at the welding speed of 100 mm/min, where the joint exhibited a higher tensile strength than that of the BM and the elongation reached 80.83% of the BM. The good mechanical properties of this joint are attributed to the refined grains and high proportion of HAGBs. However, at the welding speeds of 150 and 200 mm/min, the fracture occurred in the centre of the SZ owing to the oxide layer remnants present in the weld, and the tensile strength reached 85% and 95% of the BM, respectively. In addition, the hardness of the SZ of all joints was much higher than that of the BM owing to the fine equiaxed grains.Because the heat rapidly diffused in the ultra-thin weld, the grain size increased while the proportion of LAGBs decreased from the top to the bottom of the joint, owing to the weaker deformation at the bottom (with low hardness) than that at the top (with much higher hardness).A welding speed increase from 100 to 150 and 200 mm/min caused the grain size and the proportion of LAGBs to increase owing to the reduction in dynamic recrystallization, which was caused by the decreased heat input and strain rate.

## Figures and Tables

**Figure 1 materials-12-03906-f001:**
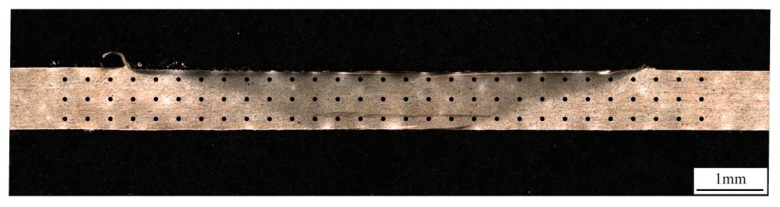
Optical microscope image of the hardness testing point distribution diagram.

**Figure 2 materials-12-03906-f002:**
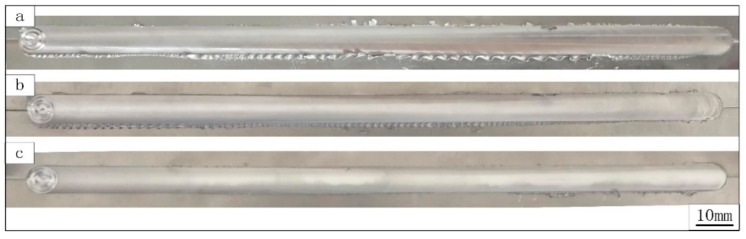
Surface appearances of the welds produced at a welding speed of (**a**) 100, (**b**) 150, and (**c**) 200 mm/min.

**Figure 3 materials-12-03906-f003:**
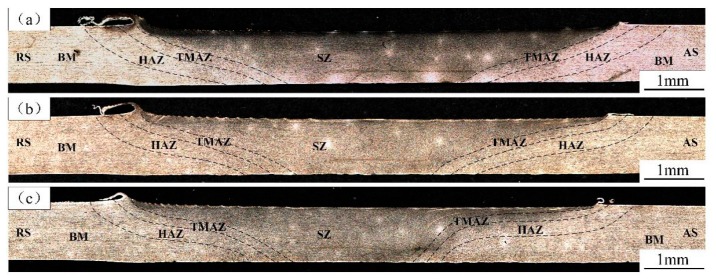
Cross-section optical macrographs of the joints produced at a welding speed of (**a**) 100, (**b**) 150, and (**c**) 200 mm/min.

**Figure 4 materials-12-03906-f004:**
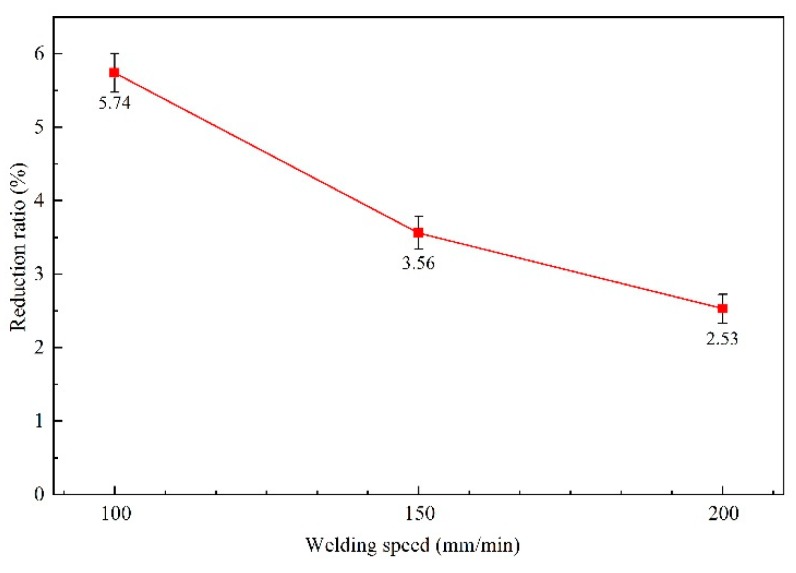
Thickness reduction rate of the joints as a function of welding speed.

**Figure 5 materials-12-03906-f005:**
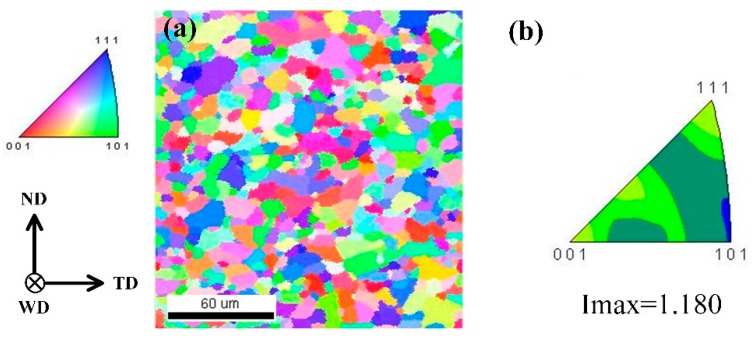
Electron backscatter diffraction (EBSD) results of the base material (BM): (**a**) inverse pole figure color map; (**b**) key of inverse pole figure color.

**Figure 6 materials-12-03906-f006:**
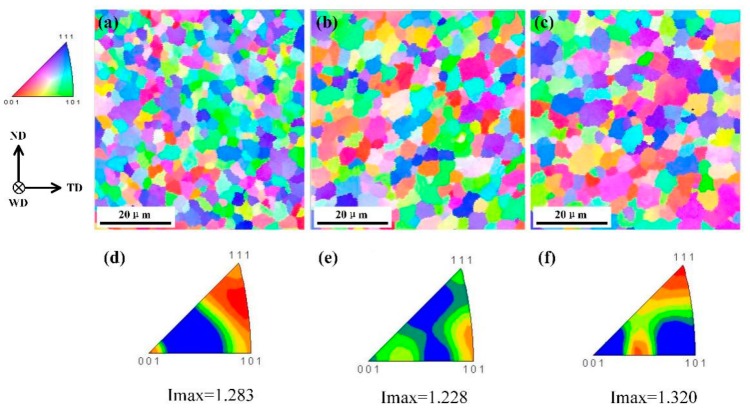
Inverse pole figure color maps of the stir zone (SZ) at the welding speed of (**a**) 100, (**b**) 150, and (**c**) 200 mm/min; and the corresponding contoured color key at the welding speed of (**d**) 100, (**e**) 150, and (**f**) 200 mm/min.

**Figure 7 materials-12-03906-f007:**
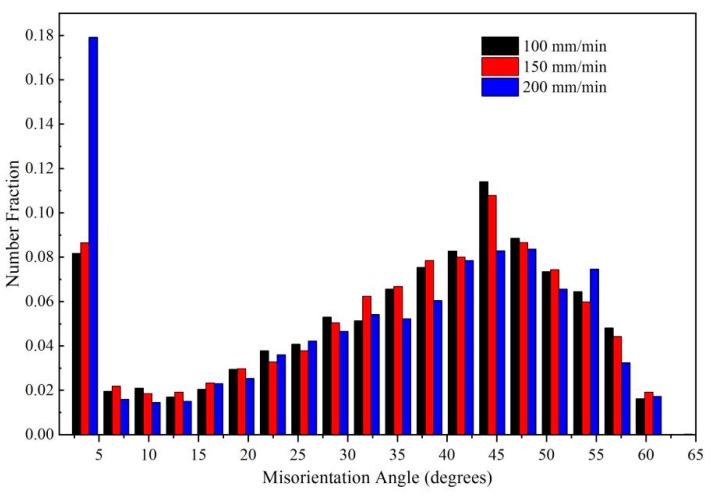
Distribution of the misorientation angles in the SZ of welds produced at different welding speeds.

**Figure 8 materials-12-03906-f008:**
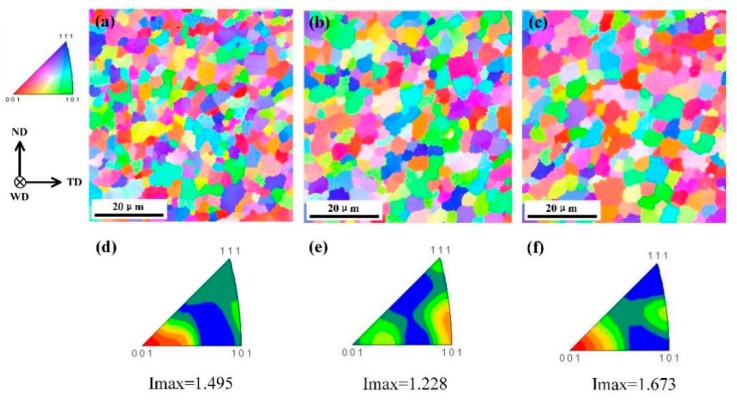
Inverse pole figure color maps of the (**a**) top, (**b**) centre, and (**c**) bottom of the SZ; and inverse pole figures of the (**d**) top, (**e**) centre, and (**f**) bottom of the SZ of the weld produced at a welding speed of 150 mm/min.

**Figure 9 materials-12-03906-f009:**
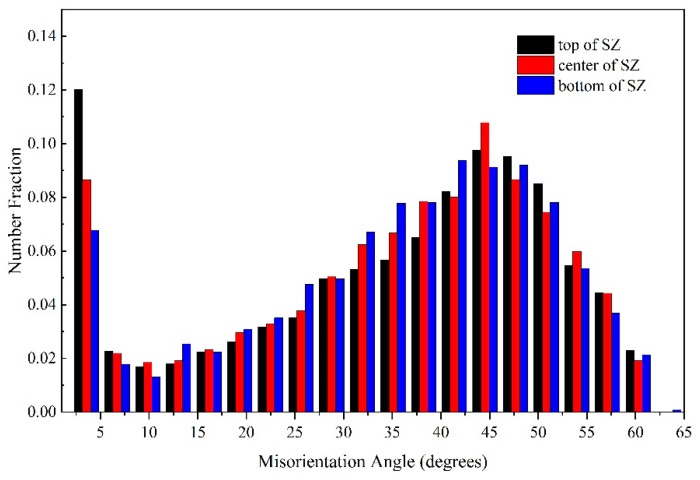
Distribution of the misorientation angles within the SZ of the weld produced at a welding speed of 150 mm/min.

**Figure 10 materials-12-03906-f010:**
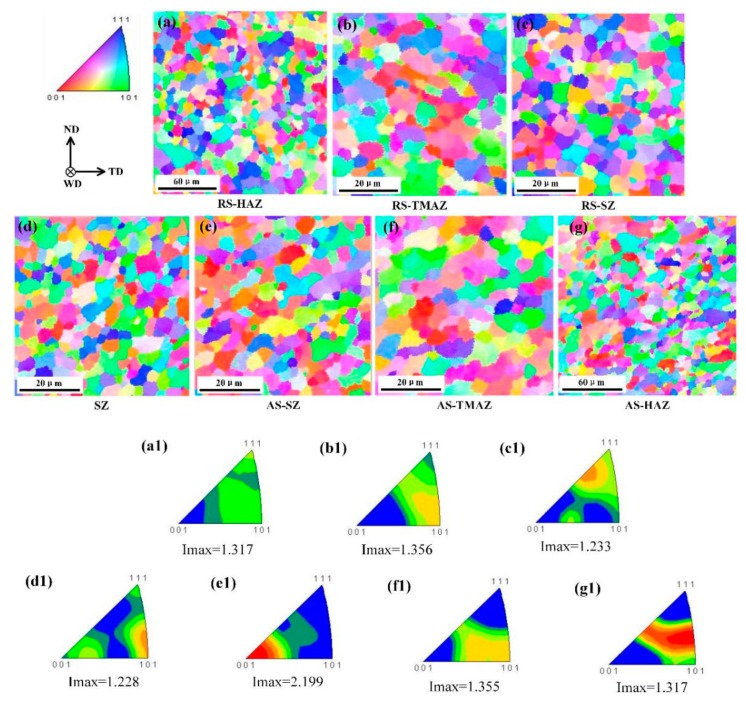
(**a**–**g**) Inverse pole figure color maps, and (**a1**–**g1**) corresponding color keys of the (**a**,**a1**) RS-HAZ, (**b**,**b1**) RS-TMAZ, (**c**,**c1**) RS-SZ, (**d**,**d1**) SZ centre, (**e**,**e1**) AS-SZ, (**f**,**f1**) AS-TMAZ, (**g**,**g1**) AS-HAZ of the weld produced at a welding speed of 150 mm/min. AS: advancing side; RS: retreating side; HAZ: heat affected zone; TMAZ: thermo-mechanically affected zone.

**Figure 11 materials-12-03906-f011:**
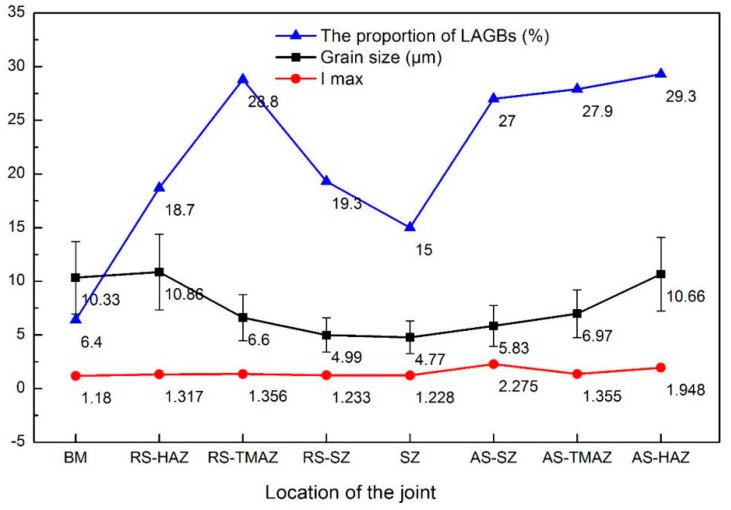
Proportion of low-angle grain boundaries (LAGBs), grain size, and maximum texture intensity of the different zones within the joint produced at a welding speed of 150 mm/min.

**Figure 12 materials-12-03906-f012:**
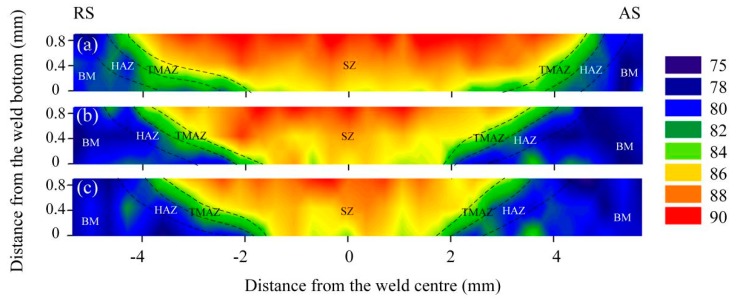
Hardness maps of the welds produced at a welding speed of (**a**) 100, (**b**) 150, and (**c**) 200 mm/min.

**Figure 13 materials-12-03906-f013:**
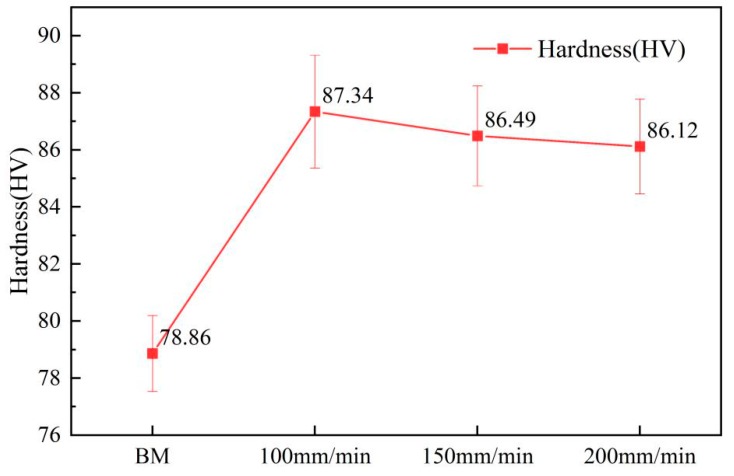
Average hardness of the BM and the joints produced at various welding speeds.

**Figure 14 materials-12-03906-f014:**
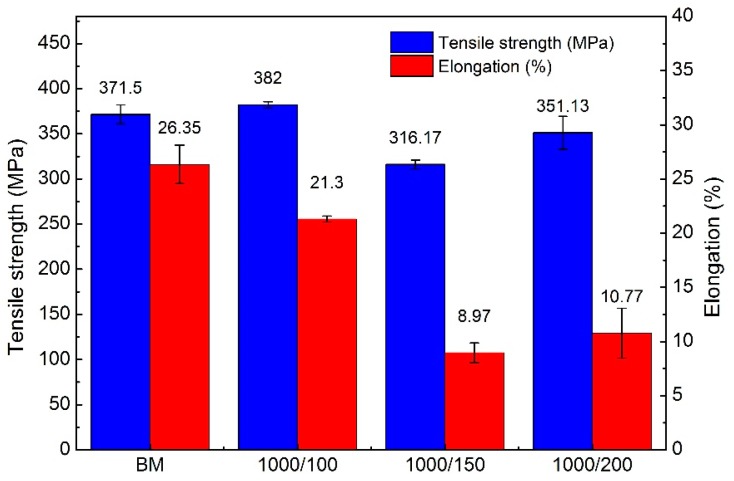
Tensile properties of the BM and the friction stir welding (FSW) joints produced with various welding speeds.

**Figure 15 materials-12-03906-f015:**
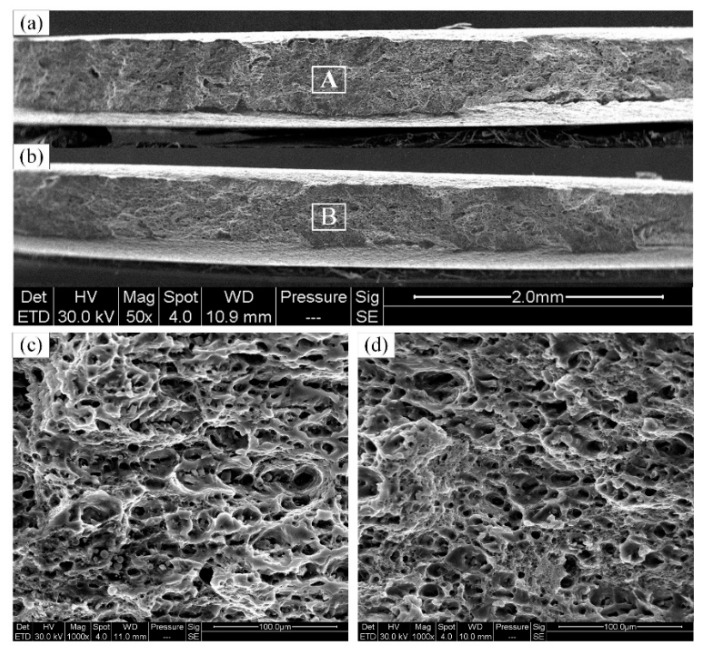
SEM images of the fracture surfaces of the (**a**) BM, and (**b**) FSW joints with welding speed of 100 mm/min, (**c**) magnified image of area A in (**a**), and (**d**) magnified image of area B in (**b**).

**Figure 16 materials-12-03906-f016:**
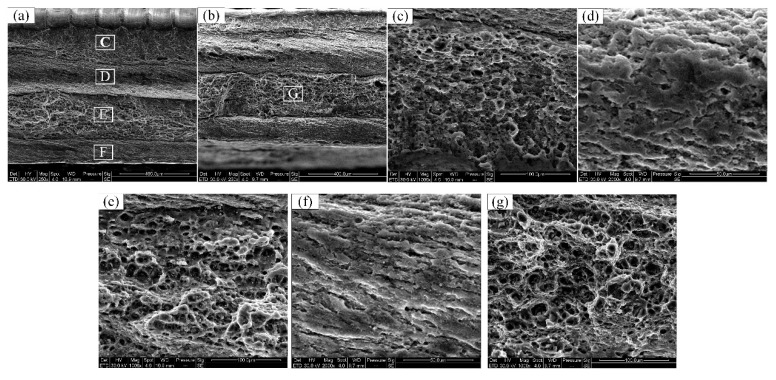
(**a**,**b**) SEM images of the fracture surfaces of FSW joints produced at a welding speed of (**a**) 150, and (**b**) 200 mm/min. (**c**–**g**) High-magnification views of the C, D, E, F, and G regions, respectively, indicated in (**a**) and (**b**).

**Table 1 materials-12-03906-t001:** The welding parameters.

No.	Welding Speed (mm/min)	Rotation Speed (rpm)	Tilt Angle (°)	Plunge Depth (mm)
1	100	1000	0.5	0.86
2	150	1000	0.5	0.86
3	200	1000	0.5	0.86

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
