# Peer review of "Microstructural Evolution and Mechanical Properties of Friction Stir Welded Butt Joints of 5A06 Alloy Ultra-Thin Sheets"

_materials, 2019, doi:10.3390/ma12233906_

Round 1

Reviewer 1 Report

The microstructure and mechanical properties of friction stir welded butt joints of 5A06 alloy ultra-thin sheet are studied. The techniques of stir welding of Al-Mg alloys and their FSW joining with other metals known very well and presented in the many papers and dissertations. Therefore, this analysis of 5A06 alloy ultra-thin sheet structure is actual only for confirmation of known facts. The review of this information presented in the introduction or chapter. However, authors do not show or claim, that microstructure of 5A06 alloy friction stir welded joint never studied before. For example, the friction stir welding technology and optimisation of welding parameters, microstructure and mechanical properties of friction stir welding of Al-Mg (5XXX) series) and exactly 5A06 alloys thick sheets are presented in the articles 10.1016/j.msea.2018.08.020, 10.1016/j.matdes.2016.10.030, 10.1111/j.1747-1567.2009.00563.x,  10.1179/136217110X12785889550181. The authors of one of this publication 10.1016/j.msea.2018.08.020 are the same as the authors of reviewed article. Therefore, the reasons of this research and review of published articles must be improved depending on the previous research results. Authors must explain more detail the main reasons of this study and importance of presented experiments results.

The chapter 2 present the information about  friction stir welding parameters, but reasons of constant and variable welding parameters selection don’t explained and clear. This selection must have better argumentation. The figure 1b is the picture of typical or standard sample. There is no big reasons to put such picture in the article. The authors write, that the all friction stir welds are well formed without any defects, especially internal defects. However, this statement do not have real proving applying X-Ray non-destructive testing, SEM or TEM analysis.  The soundness and quality of welded joint must be confirmed by NDT or another effective method. It’s hardly credible that welding parameters without preliminary studies can warrant the good results. There is now such important information in this article or correct references to previously published experimental results. Authors in 3 chapter annotate, that the decrease of joint mechanical properties of samples welded with 150 and 200 mm/min speed can be due to oxide layer in the weld. It is unclear, how the authors established the tensile strength of joint area of sample welded with 100 mm/min speed. The fracture of joint is located in basic metal, the average hardness of this joint is also highest. The authors do not show the mechanical properties (tensile strength, elongation and hardness) of base material A506. The tensile test in this case shows the properties of basic material. The statement, that the strength of joint in this case reached 102.83%, is very doubtful. The authors must add information about properties of base material in the article and extend analysis and comments of mechanical tests results.

Reviewer 2 Report

SUMMARY

The article contains the results of the analysis of microstructure changes of thin sheets of 5A06 alloy by FSW.
The description of the results is very extensive and detailed. The conclusions drawn are correct.

MAJOR COMMENTS

in my opinion the presentation of the hardness measurement results as in fig. 12 is low transparent and makes analysis difficult.
However, the description and the conclusions in this regard are correct.

I suggest marking in fig. 12 the designation of joint zones (SZ, RS-SZ, AS-SZ ...).

SUGGESTED IMPROVEMENTS

Line 73 I suggest giving the temperature in Kelvin

Line 89 is this a valid reference No[5]?

Reviewer 3 Report

Some of questions and comments include:

How did you choose the upper and lower limit for welding speed during your parameter setup and selection? Please elaborate Why did you not investigate any alternative parameters in the rotation speed, tilt angle and plunge depth? Please rebut Is welding speed the only critical parameter? "According to our previous studies, the welding conditions were chosen and shown in Table 1"? Can you please provide further data, or citations? "The deviation of grain size in the stir zone is 1.12 μm at the welding speed of 100 mm/min, while it increased to 1.51 μm and 1.66 μm at the welding speed of 150 and 160 mm/min, respectively". Based on this statement was 160 mm/min investigated? If so, why, and what additional data points were investigated? "The deviation of grain size in the stir zone is 1.12 μm at the welding speed of 100 mm/min, while it increased to 1.51 μm and 1.66 μm at the welding speed of 150 and 160 mm/min, respectively. Firstly, during the welding process, the more complete the dynamic recrystallization is, the more homogeneous the microstructure is. Secondly, during the cooling stage, the grain would grow under the high temperature conditions. At the welding speed of 100 mm/min, the high temperature dwell time is longer than that of 150 mm/min and 200 mm/min, and Luo et al. [18] concluded that grain growth is a process in which large grains grow at the expense of their small neighbors. So that longer high temperature dwell time after welding is beneficial to the growth of grains by the expense of smaller grains, at the same time it would lead to the reduction of grain size deviation." Isnt the data you obtained from EBSD contradicting the theory of grain size deviation from the citation? Based on this theory and your explanation, shouldn't your data also be reflecting reduction in the grain size deviation? In your case, the grain size is increasing, but the deviation is also increasing. "The texture intensity of the SZ is slightly stronger than that of the BM under all welding parameters, which may be due to the fact that the heat input and deformation did not reach the condition of complete dynamic recrystallization under the welding parameters used in this study, and the regular stirring action of pin also led to the concentration of grain orientation in some directions." This again goes back to the question of how and why the parameter values and their limits were chosen? What is the target? "According to the results in Figure 6 & 7, it could be concluded that the mechanical properties of the weld with 100 mm/min was the best." Please be careful with generic statements like this. While I understand your implication in theorizing that small grain sizes and specific orientations can yield some good mechanical properties, you cannot "conclude" of the same without providing more evidence and clarifying which specific property you are referring to. For example, based on the application; good strength, good elongation, good hardness, good impact toughness, good fatigue life are each good mechanical properties. But some of them cannot be achieved without forgoing another. Some clarity on the targeted properties and what makes them good would be appreciated.  "During the FSW of ultra-thin aluminium plate, the effect of temperature gratitude along vertical direction of SZ can be ignored due to the good thermal conductivity and thin thickness of ultra-thin aluminium plate." I think you mean gradient. Please correct or rebut. Are the authors implying that the fracture and tensile/elongation performance of 200mm/min vs 150 mm/min is due to there being a smaller oxide layer and delamination (from reference) in the 200 mm/min? Why does this happen? Can you please elaborate on this causality or rebut and clarify why 200 mm/min exhibits higher strength and elongation than 150 mm/min? From what I can tell, the increase in strength between BM and 100 mm/min is very small (is it even statistically significant?) while the decrease in elongation between the two is very large. Is this the end goal of the research? Are those the properties that are being targeted? Please clarify and elaborate on why this is important. "Besides, the average hardness of the weld is 87.34 HV, which is much higher than that of base metal." What is the hardness of the base metal? I only found the hardness of the three weld experiments. Can you please put this in a table or image similar to the tensile and elongation chart.

Reviewer 4 Report

This is much better than the previous version of this report. The authors addressed issues raised by the reviewers previously. However, the introduction needs improvement explaining the novelty of the current study. I would suggest review few other joining technology such as clinching, self piercing riveting and resistance spot welding. The following paper would give a good understanding:

Mechanical joining of various materials by clinching method

Quality of self-piercing riveting (SPR) joints from cross-sectional perspective: A review

Review of dissimilar friction stir welding between titanium and aluminum

A review on resistance spot welding of aluminum alloys

Author Response

Dear reviewer,

I am sorry that there may be some little errors. We haven't submitted the revised version of the paper. Your review opinions are inconsistent with our paper, it may be from another revised article.

Best wishes to you!

Round 2

Reviewer 1 Report

This draft of article is much better. Authors add the important information, argumentation and deeper analysis, correct the mistakes and incorrect expressions. Answers to all reviewer questions and  remarks are acceptable and comprehensive. 

Author Response

Thank you  very much!

Reviewer 3 Report

I have some comments based on past observations and current revisions:

While I understand that microstructural evolution was one of the aspects of the investigation, wouldn't varying another parameter also affect microstructural evolution? By the authors' own admission, rotation speed does affect the weld significantly and has an upper and lower bound. By only changing one parameter, I cannot help but wonder if I am being forced to draw conclusions by only looking at part of the process behavior and mechanics. I have found other published work that pursue comprehensive experimental investigations and microstructural work. In this case, what is the novelty and/or the importance of this specific parameter that the authors' are postulating?

Parameters also interact and have an effect on one another which cannot be observed in this type of smaller experimental design. Hypothetically, if I were to change one of the other parameters the authors have specified, would the microstructural evolution (and thereby the mechanical properties) change significantly? If that is the case, as an independent researcher, I am not certain I can use the experimental data offered by the authors as a guideline to pursue tangential investigations.

From my observation of the charts, the strength (plus margin) of the 100mm/min welds fall within the range of the base metal's strength (plus margin). I am not certain that there is a statistically significant difference in the strength of the two. Can the authors provide additional data or analyses to support their claim of "higher" strength?

Reviewer 4 Report

The introduction needs improvement explaining the novelty of the current study. I would suggest review few other joining technologies such as clinching, self-piercing riveting and resistance spot welding. The following paper would give a good understanding:

Mechanical joining of various materials by clinching method

Quality of self-piercing riveting (SPR) joints from cross-sectional perspective: A review

Review of dissimilar friction stir welding between titanium and aluminum

A review on resistance spot welding of aluminum alloys

Additionally, it is not clear how the upper and lower limit of the speed of welding was decided. The authors need to explain why the strength of 200mm/min is higher than 150 mm/min. What is the hardness of base metal as you stated that hardness of the weld is high? The relation between the welding speed and grain size is vague and not well explained.

Round 3

Reviewer 4 Report

Can be accepted now.

Author Response

Dear reviewer,

Thank you again for the comments about our paper submitted to Materials (materials-607994). The manuscript have been checked by a professional English editing service, and the revised version have been submitted with the main change highlighted in red.

Best wishes!